# Validation of Rayleigh Wave Theoretical Formulation with Single-Station Rotational Records of Mine Tremors in Lower Silesian Copper Basin

**DOI:** 10.3390/s21103566

**Published:** 2021-05-20

**Authors:** Witold Pytel, Krzysztof Fuławka, Piotr Mertuszka, Bogumiła Pałac-Walko

**Affiliations:** 1Research & Development Centre, KGHM Cuprum Ltd., 2-8 Sikorskiego Street, 53-659 Wrocław, Poland; wpytel@cuprum.wroc.pl (W.P.); kfulawka@cuprum.wroc.pl (K.F.); pmertuszka@cuprum.wroc.pl (P.M.); 2Faculty of Geoengineering, Mining and Geology, Wrocław University of Science and Technology, 15 Na Grobli St., 50-421 Wrocław, Poland

**Keywords:** rotational seismology, Rayleigh wave, rotational seismic load

## Abstract

The classical Rayleigh surface rotational wave in terms of its theoretical notation and, resulting from this, properties associated with the induced seismic phenomena in mines are presented. This kind of seismic wave was analysed in-depth from the point of view of the parameters governing the form of its mathematical notation based on the similarity to the records obtained during the induced seismicity in near-field 6-DoF monitoring. Furthermore, conducted field measurements made it possible to relate the amount of the emitted seismic energy to the expected highest amplitude of rotational vibrations in the entire field of their impact on the rock mass. As a result, this made it possible to impose the completely defined R wave to the numerical models of given objects; the safety level, when subjected to the dynamic load induced by the rotational wave, would be an objective of the performed analyses. The conducted preliminary analyses were prepared for a plane strain state, for which the values of seismic rotations were evaluated concerning the energy and the distance of the seismic event’s source. As a result of the performed simulations, it was found that the results of the calculations matched with a satisfying degree with the field seismic measurements of the rotational ground motion induced by propagating the seismic wave. Such a verified analytical description of the theoretical formulas can be the basis for the implementation of R-wave characteristics into seismic codes and numerical analyses of object stability in the Lower Silesian Copper Basin region.

## 1. Introduction

Seismic activity is one of the deadliest and destructive source of hazards affecting both the local society and the environment. In general, seismic events may be divided into natural and anthropogenic ones [1,2,3,4]. Natural earthquakes occur due to a sudden release of energy accumulated in Earth’s crust, usually when the masses of rock are acting mutually against one another, which causes sudden fracture and slip along the fault lines. In turn, anthropogenic quakes may be defined as seismic events caused by human activities [5,6] such as mining [7,8], groundwater extraction [9], fluid injections [10,11] dam construction [12], firing explosives [13] and nuclear explosions [14]. Regardless of the type of seismic event, the additional dynamic load may generate a significant damage to structures located near the seismicity source. As it was pointed out by Albano et al. [3], with the rise of the intensity of anthropogenic earthquakes, an increased number of unexpected damages is being observed [8,15,16,17]. Many of them are directly related to seismic load occurrences that caused a lot of tragic social, economic and environmental loses [18,19]. This is particularly visible in the regions of mining-induced seismicity [20,21]. Special attention should be paid to tailing storage facilities (TSF), which are currently one of the largest and most environmentally dangerous engineering structures, often located in the direct vicinity of mines due to economic reasons. As it was pointed out by Owen et al. [22], these facilities are currently characterized by one of the highest failure rates among all engineering structures [23,24]. According to the International Commission on Large Dams report, in the 21st century, at least once a year, a disaster related to the loss of stability of TSF slopes has been observed [25,26]. Such a situation may be related, among others, to an incomplete description of the seismic load.

Besides that, in most of the available seismic codes and numerical methods, the seismic load is defined as a clear translational ground motion, which describes the characteristic of Primary (P) and Secondary (S) body wave propagations through the ground. At the same time, the rotational load, generated mostly by Rayleigh (R) and Love (L) surface waves, as well as with body wave interactions with the ground surface [27], are neglected, mainly due to the lack of rotational measurements of the ground motion [28,29,30]. The significant development in six degrees of freedom (6-DoF) measurements of earthquakes and mining induced tremors has been noticed in the last 10 years when numerous essential research works have been reported in the field of development of new systems for integrated measurements of rotational and translational components of the ground motion [31,32,33,34,35,36]. However, still, regardless of the extended research and availability of 6-DoF measuring systems, so far, there has been no significant development in the methods of numerical or analytical modeling of a slope stability prediction.

In the case of earth dams, the problem of their stability under seismic load conditions is solved with use of the Newmark method [37], which assumes that, whenever the seismic acceleration is higher than the slope critical acceleration [38], causing collapse, then permanent displacements occur. The amount of these displacements may be obtained by integrating twice, the difference between the applied acceleration and the critical acceleration with respect to time [39]. However, this approach is based solely on the translational components of seismic waves. This way, the rotational seismic movements related mostly to the surface wave propagations are neglected [40,41]. Therefore, there is a strong need for verification if the mathematical description of the rotational ground motion developed so far fits the real field-measured data. If so, further improvements within slope stability methods, with use of 6-DoF motion, could be elaborated.

Within this paper, the theoretical notation of the Rayleigh wave characteristic for the conditions of the Lower Silesian Copper Basin area are presented and verified afterward with the data obtained with continuous single-station 6-DoF monitoring at the west slope of Zelazny Most Tailing Pond. The Rayleigh wave characteristics have been separated from the velocity records using the method of time–frequency decomposition. Finally, a collected database of 6-DoF records was used for determining the relation between the rotational ground movement, seismic energy and vibration frequency of the tremor.

## 2. Materials and Methods

Continuous 6-DoF measurements in the LSCB region have been conducted in the Rudna mining area, which, at the moment, is characterized by the highest seismicity among all three KGHM copper mines. For the purposes of this paper, 6-DoF records of three tremors recorded in the Rudna mining area have been used. The first two records were characterized by high-energy reaching (E = 1.1 × 10^7^ J and E = 3.1 × 10^8^ J), while the third was categorized as a seismic event with moderate energy (E = 9.7 × 10^5^ J). The locations of the tremor epicenters and the measuring site are presented in Figure 1.

When analyzing Figure 1, it may be observed that the wavefront of each analyzed tremors is not oriented strictly along the X- or Y-axis; therefore, the actual values of the velocity and seismic rotation may slightly differ from the recorded ones. In a case of far-field earthquakes, the procedure of amplitude correction according to the azimuth of the tremors may proceed; however, in the case of near-field high-energy tremors, using such an approach may lead to significant errors. This is because the tremors of high energies (in LSCB conditions) in general are related to the slip on the fault surface. Since this surface may be a few hundred meters in width, the exact spatial location of the source based on arrival times may be assessed incorrectly. Knowing that the deviation of the X and Y coordinate determinations may reach up to a few hundred meters, and also bearing in mind that numerous tremors occur within 2–8 km from the area of interest, it may be concluded that the determined azimuth may be burdened with an error of several degrees. This is why in the research presented here in the procedure of azimuth correction was not utilized.

The data presented in Table 1, including these event rotation amplitudes, as well as their frequency distributions, have been used in the mathematical description of the Rayleigh wave characteristics.

The measuring station was installed in a 2-m-deep concrete well, located at the dam of Zelazny Most Tailing Pond. For the measurements of translational ground motion, a EP-300 seismometer was utilized, while the rotational movements were recorded with the use of R-1-rotational seismometers. The 6-DoF data were collected with a DR-4050P 24-bit seismic recorder, supplied with a 32-GB removable memory (Figure 2). Both seismometers as well as seismic recorder were manufactured by Eentec company, Kirkwood, MO, USA, sampling rate of 500 Hz was used during the conducted measurements.

Y-axes of both translational and rotational seismometers were directed into the north direction, while the X-axes were facing the west. The rotational seismometer, due to its low weight, was grouted to the floor to ensure proper measurements. More details about the measuring system were presented in reference [42].

## 3. Results

### 3.1. Rayleigh Wave Mathematical form Assessment

The principal dynamic problem of the theory of linear elasticity may be presented as the system of three differential equations of motion (Navier’s equations) engaging three elastic parameters of the body and its density parameter. Since the final goal of this analysis is to determine preliminarily the influence of the surface wave transition on the stability of embankments and earth dams working in a plane deformation state, a system of two Navier equations, representing mathematically a plane wave traveling in an isotropic infinite elastic body, should be solved. This way, the following two equations of waves translational motion can be obtained:(1a)$$u_{P}=A\sin\left[\frac{2\pi }{\lambda_{P}}\left(x-v_{P}t\right)\right]$$(1b)$$w_{S}=A\sin\left[\frac{2\pi }{\lambda_{S}}\left(x-v_{S}t\right)\right]$$where *u_p_* is the primary (longitudinal) wave, *w_s_* is the secondary (transverse) wave, $\lambda_{P}$ and $v_{P}$ are the wavelengths (m) and velocities (m/s) of a primary wave, respectively, $\lambda_{S}$ and $v_{s}$ are the wavelength (m) and speed (m/s) of a secondary wave and $\frac{2\pi }{\lambda_{S}}=\kappa_{s}$ and $\frac{2\pi }{\lambda_{p}}=\kappa_{p}$ are the wave numbers.

The primary and secondary waves reaching the body surface are reflected and refracted, which may lead to their special combination, resulting in surface rotational Rayleigh waves (R-waves) generating.

The Rayleigh wave mathematical form of notation, which is a basic problem solution, has been exhaustively analyzed by Kolsky [43] and Fung [44], who used the potential function approach. This approach also has great potential in terms of its possible implementation into numerical codes. Therefore, within the presented analysis, the abovementioned methodology is used and validated with in situ 6-DoF measurements.

The first stage of the analysis involves the tremor of the energy of E = 3.1 × 10^8^ J. For the purpose of further calculations, the acceleration of translational and rotational motion was determined. Waveforms representing the rotational and translational components of the seismic wave generated by this tremor with their spectral characteristics are presented in Figure 3.

Since the methodology presented within the framework of this analysis was developed in two dimensions, it was necessary to determine which component of the seismic wave should be chosen as a reference system. Since, according to field measurements, the rotation noticed around the Y-axis (N–S direction) reached the highest value, the amplitude data from the Y-rotational component was used in further analyses.

In the next steps, there were made the attempts for defining the R-wave theoretical characteristics based on the recorded waveform. This goal seemed to be especially challenging in the near-wave field case, due to small distances from the seismic event sources. According to Maranò and Fäh [45], the surface waves, due to the characteristics of their propagation, are not fully identifiable with the use of a single three-component translational sensor. Still, these waves may be identified with the use of three-component rotational seismometers. As it was pointed out by Yan et al. [46], horizontal components of seismic rotations may be useful for the determination of the Rayleigh characteristics, which propagate through the surface, causing elliptical motions of ground particles. In turn, the vertical (Z) component of the seismic rotation reflects the Love wave propagation characteristics, which is polarized horizontally, perpendicular to the direction of the propagation.

Therefore, from the point of view of the research presented herein, it was concluded that horizontal components of rotational seismic motion will be particularly useful for the purpose of Rayleigh wave extraction from seismic records.

In order to separate surface waves from the overall ground motion, the qualitative methodology of the time–frequency decomposition was used. A sophisticated algorithm related to this topic was presented in the paper by Sollberger et al. [47,48], where a time–frequency decomposed representation of the 2018 Gulf of Alaska earthquake by computing the S-transform was performed. This approach seems to be very promising in terms of the automatic detection of surface waves, but at this stage, it has not been verified yet in the LSCB geological mining conditions, where the epicentral distance of the source of the tremors is often lower than 7 km. Therefore, within the presented paper, the time–frequency characteristic and separation of the R-wave characteristics were performed manually. The results of the analysis are presented in Figure 4.

For the determination of the R-wave arrival time, the rotation about the horizontal axes was analysed. Based on the theory concerned with the seismic body and surface waves characteristics, one may conclude that R-waves propagate with the lower velocities and generate ground vibrations of the lower frequencies in comparison with the body wave motion effects. Thus, when analyzing seismic records, it may be observed, especially with respect to the horizontal components, that a significant drop in the values of the dominant frequency and accumulation of the seismic energy occurred between the 39th and 41st seconds of the time series. On this basis, it was assumed that this part of the waveform represents the R-wave arrival and its effects in the measuring site’s neighborhood. The R-wave’s separated velocity and acceleration signals, both in the time and frequency domain, are presented in Figure 5.

The spectral characteristics of the recorded R-wave indicates that its dominant frequency is about *f* = 6 Hz. Therefore, in further calculations, a sinusoidal wave of the frequency 6 Hz, propagating in the x–z planes with the velocity of $v_{R}$ along the x-axis and rotating around the y-axis, is under consideration herein.

Considering the velocity of the seismic waves propagation, it have to be emphasized that, in the LSCB area, the value of the R-wave propagation velocity was determined indirectly based on the measured S-wave velocities. This is mostly due to the lack of in situ measurements concerning the propagation velocities of the surface waves in the near-field induced by mining activity in the LSCB area. According to Achenbach [49], the Rayleigh wave velocity is generally about 10% lower than the velocity of the S-wave. The estimation of theR-wave velocity for the purposes of the present research is presented below.

According to Rayleigh’s solution, the horizontal and vertical displacements of the point within the elastic half-space triggered by the Rayleigh plane wave may be expressed by the following relationships:(2)$$u=A\kappa \left[e^{-qz}-2qs{\left(s^{2}+\kappa^{2}\right)}^{-1}e^{-sz}\right]\;\sin\kappa \left(x-v_{R}t\right)$$
(3)$$w=Aq\left[-e^{-qz}+2\kappa^{2}{\left(s^{2}+\kappa^{2}\right)}^{-1}e^{-sz}\right]\;\cos\kappa \left(x-v_{R}t\right)$$
where $\kappa =\frac{2\pi }{\lambda_{R}}=\frac{2\pi f}{v_{R}}=\frac{\omega }{v_{R}}$ is the wavenumber, *q* and *s* are the constants, *x* and *z* are the horizontal and vertical coordinates of the Cartesian system and *t* is time coordinate. The value of $v_{R}=\kappa_{1}v_{S}$ may be determined by solving the characteristic equation of Rayleigh:(4)$$\kappa_{1}^{6}-8\kappa_{1}^{4}+\left(24-16\alpha_{1}^{4}\right)\kappa_{1}^{2}+\left(16\alpha_{1}^{2}-16\right)=0$$
where $\alpha_{1}^{2}=\frac{1-2\nu }{2\left(1-\nu \right)}={\left(\frac{v_{S}}{v_{P}}\right)}^{2}$, assuming the value of $v_{S}$ (Table 2).

The values of *κ*_1_ may be found also directly in Figure 6 and Table 3.

Assuming *ν* = 1/3, one may get $\alpha_{1}^{2}=1/4$, and then, from Equation (4), we have
(5)$$\kappa_{1}^{6}-8\kappa_{1}^{4}+\left(24-16\alpha_{1}^{2}\right)\kappa_{1}^{2}+\left(16\alpha_{1}^{2}-16\right)=\kappa_{1}^{6}-8\kappa_{1}^{4}+20\kappa_{1}^{2}-12=0$$
where, from one real root of the equation that may be estimated as $\kappa_{1}^{2}=0.8696$, $\kappa_{1}=0.9325$.

This means that the surface wave R in the analyzed conditions travels with a velocity of 93.25% of the propagation of the secondary wave S. Based on that, one may already assess the following parameters of Equations (2) and (3):(6)$$q=\kappa \sqrt{1-\alpha_{1}^{2}\kappa_{1}^{2}}\;s=\kappa \sqrt{1-\kappa_{1}^{2}}$$
where q=0.8847κ and s=0.3623κ. In the next step, based on reference [50], it is assumed that, for granular geological formations located closely to the surface in close vicinity to Zelazny Most Tailing Pond, and knowing that the velocity of the shear wave in this area is close to $v_{S}\;$= 320 m/s [52], it is justified to estimate its velocity as $v_{R}$ ≅ 0.3 km/s (see, also, Table 2). However, it is worth noticing that this estimate has an approximate character, and it may be recommended only in the preliminary stage of the investigation. As they progress, it is recommended to define it more precisely based on the relevant field measurements.

From the notation of Equation (2), one may conclude that its element
(7)$$e^{-qz}-2qs{\left(s^{2}+\kappa^{2}\right)}^{-1}e^{-sz}$$
is a measure of the horizontal displacement u_x_ attenuation rate with the depth of *z*. If one substitutes for the above equation the calculated values of *q* and *s*, one can get the following numerical factor:(8)$$e^{-0.8847\kappa z}-0.5667\;e^{-0.3623\kappa z}$$
which value tends to zero at the normalized depth of *κ**z* = 1.091 (Figure 7, left). Taking into account that wave number $\kappa =\frac{2\pi f}{v_{r}}=\frac{2\pi }{\lambda_{R}}=0.1257$, the assessed depth corresponds to the depth of $z=0.174\lambda_{R}$. On the other hand, since *f* = 6 Hz, as well as the velocity of the wave, the R propagation is estimated as *v_R_* = 300 m/s, and the attenuation depth for the horizontal component of the Rayleigh wave displacement is *z* = 8.69 m below the ground. In turn, the attenuation depth normalized with respect to *κ* is presented as the function of Poisson’s ratio values (Figure 7, right).

Doing similarly with respect to the vertical component *w* of the *R* wave, the element of Equation (6) that governs the rate of decay of this component with the depth was selected as follows:(9)$$-e^{-qz}+2\kappa^{2}{\left(s^{2}+\kappa^{2}\right)}^{-1}e^{-sz}$$

Substituting the appropriate numerical values of the parameters into Equation (9), the following function of decay is obtained:(10)$$-0.8847e^{-0.8847\kappa z}+1.5641\;e^{-0.3623\kappa z}$$
which value reaches the maximum at a depth of $z_{0}=0.098\;\lambda_{R}=4.9$ m; after this, without changing its sign, it monotonically tends to zero (see Figure 7, left).

Finally, the equations describing the horizontal and vertical components of the R-wave take the following form:(11)$$u\left(x,z,t\right)=A_{1}\kappa \left[\exp\left(-qz\right)-2qs{\left(s^{2}+\kappa^{2}\right)}^{-1}\exp\left(-sz\right)\right]\;\sin\left[\;\kappa \left(x-v_{R}t\right)\right]$$
(12)$$w\left(x,z,t\right)=A_{2}q\left[-\exp\left(-qz\right)+2\kappa^{2}{\left(s^{2}+\kappa^{2}\right)}^{-1}\exp\left(-sz\right)\right]\cos[\kappa \left(x-v_{R}t\right)]\;$$
where $\kappa =\frac{2\pi }{\lambda_{R}}=0.1256$, q=0.8847κ, s=0.3623 κ, $v_{R}=300$ m/s is the velocity of the R-wave propagation in granular and poorly consolidated glacial deposits and *A_1_* and *A_2_* are the numerical coefficients, allowing to reduce the calculated displacements to the amplitudes of wave motion components in the horizontal direction along the x-axis and in the vertical direction along the z-axis, measured under field conditions.

Taking into account the above-mentioned numerical data, one may obtain the following notations for the displacements of the medium loaded with the R-wave transition:(13)$$u\left(x,z,t\right)=A_{1}\;0.1256\;\left[\exp\left(-0.1111z\right)-0.5667\exp\left(-0.0455z\right)\right]\sin\left[0.1256\;\left(x-300t\right)\right]$$
(14)$$w\left(x,z,t\right)=A_{2}\;0.1256\left[-0.8847\exp\left(-0.1111z\right)+1.5641\exp\left(-0.0455z\right)\right]\cos[0.1256\left(x-300t\right)]$$

The calculated instantaneous displacements of the model surface (*z* = 0 m) for *t* = 5 s are shown in Figure 8. They were normalized with respect to the maximum values of the vibration amplitudes *A_1_* and *A_2_*, which, at the present stage of the problem knowledge, are not known yet.

One may notice that, on the surface, $u_{extr}=\pm 0.0545\;A_{1}$ as well as $w_{extr}=\pm 0.0854\;A_{2}$; thus, $\left|\frac{w_{max}}{u_{max}}\right|=1.56\;\frac{A_{2}}{A_{1}}.$ Derivatives of the displacement functions, such as the horizontal strain on the surface, $\varepsilon_{x}=\frac{\partial u\left(x,z,t\right)}{\partial x}$ reaches the extremal values of $\varepsilon_{extr}=\pm 0.0066\;A_{1}$ and the surface tilt or, in other words, the surface rotation around the y-axis, $T_{y}=\frac{\partial w\left(x,y,t\right)}{\partial x}$, which values remain within the limits of $T_{y,extr}=\pm 0.0107\;A_{2}\;rad$ and is characterized in Figure 8.

Furthermore, the notations of Equations (11)–(14) indicate that any particle in the medium is moving retrograde along an ellipse, which the major axis is normal to the surface. For particles located directly on the surface, the ratio of the major to the minor axes is equal to 1.56 (Figure 9). For other values of the Poisson’s ratio (Figure 10), the value of this relation remains within the limits of 1.272 ÷ 1.839.

It can be noticed that the total increase in the value of the ratio *a/b* within the possible values of Poisson’s ratio *ν* = 0 ÷ 0.5 reached 44.4%; however, the increase in the value of the vertical displacements *w* was much greater (72%) than the horizontal displacements *u*, for which the values increased by no more than 19%.

From the form of the first derivative of Equation (14) with respect to x, the mutual relationships between the three basic parameters $R_{y,max},$ *f* and $v_{R}$ (maximum amplitude of rotation of the elastic half plane’s surface, dominant wave frequency and the wave velocity, respectively) describing the tilt of the instantaneous wave surface were determined and presented in Figure 11.

It is interesting that these relationships indicate that, with a R-wave dominant frequency increment, the maximum value of the rotation also increases for all the wave velocities. Furthermore, for higher values of wave velocities, one may observe a significant decrement of the maximum value of *R_y_,_max_* rotation on the ground surface. This suggests that any structure resting on the elastic half-plane, when exposed to the seismic surface R-wave, must withstand much greater forced rotational displacements when founded on a weak/soft ground than when resting on more competent types of geological formations.

On this basis, one can initially assume that the presence of fragmented materials, such as various types of soil on the ground surface, favors the generation of larger amplitudes of the displacement components associated with the propagation of the R-wave than the presence of hard rocks, e.g., sandstone, characterized by small values of the lateral expansion coefficient. However, in the issue of the effect of Poisson’s ratio on the size of the Rayleigh wave amplitude, the problem should be considered more broadly, taking into account such material features as the velocity of the seismic wave propagation through the given media, as well as the dominant frequency, since they have a significant impact on the values of the parameters *κ*, *q* and *s* governing the R-wave equation.

Figure 5, right shows the instantaneous values of the terrain tilt *T*, which, by definition, can be identified with the original rotation measurements carried out with rotational seismometers stabilized on the ground surface, assuming, however, that the angles ϕ of the rotation around the y-axis of the terrain surface are relatively close to zero. Figure 5 justifies this assumption, since $T_{max}≅\phi_{y,max}=10.7\;\mathrm{mrad}=0.31\;\deg$.

The acceleration of the Rayleigh wave rotation (rad/s^2^) in a plane deformation state has the form
(15)$$a_{x}=0\;a_{z}=0$$
(16)$$a_{y}=\frac{\partial^{3}w_{z}}{\partial x\partial t^{2}}=-A_{2}\kappa^{2}{\left(\kappa v_{R}\right)}^{2}0.8847\;\left[-\exp\left(-qz\right)+2\kappa^{2}{\left(s^{2}+\kappa^{2}\right)}^{-1}\exp\left(-sz\right)\right]\cos[\kappa \left(x-v_{R}t\right)]$$

From the form of Equation (16), one may conclude that the maximum values of the rotational velocity and accelerations theoretically remain in a constant proportion, respectively:(17)$$\Delta_{v}=\kappa v_{R\;}=2\pi f\;\mathrm{and}\;\Delta_{a}={\left(\kappa v_{R}\right)}^{2}={\left(2\pi f\right)}^{2}$$
in relation to the excited rotations along the x-axis $\phi_{w}=\frac{\partial_{w}}{\partial_{x}}$. For the case under consideration: $\Delta_{v}=37.7$ and $\Delta_{a}=1421.3$. It is, of course, desirable that the dominant frequencies of the R wave measured in situ be consistent with the theoretical approach.

Knowing that the acceleration, which is implemented into a numerical model, is the derivative of the velocity vector as a function of time, proper values of the rotational acceleration are required. To verify if the calculated accelerations are reliable, a comparison between the rotation (ϕ), rotational velocity ($v_{y}$) and rotational acceleration ($a_{y}$) was conducted. The results of the comparison, presented in Figure 12, are based on the rotation velocity field measurements in the Rudna mining region.

One may notice that, for the dominant frequency, *f* = 6 Hz, Δ*v* = 34.1 and *Δa* = 34.1 × 45.1 = 1537.9. These increments agree satisfactorily with the theoretical values equal to Δ*v* = 37.7 and Δ*a* = 1421, respectively.

However, when examining the results presented in Figure 12, one may conclude that the values of the rotation are 12 up to 60 times less than the rotational velocity values, generally remaining in a linear function of the dominant frequency of the seismic wave. A similar conclusion may be drawn by analyzing the dependence between rotational velocity and rotational acceleration. In this case, the acceleration values are from 16 up to 88 times greater than the values of the rotation velocity. From a theoretical point of view, both curves should overlap. However, some inaccuracy may be generated both due to the measurement process and due to data processing. According to the manufacturer, the frequency band standard in the R-1 rotational seismometer is in the range of 0.05 to 20 Hz. Since, in the recorded waveforms, high frequencies occurred, the seismic signal was filtered with the use of a bandpass filter in the range of 0.5–20 Hz. Then, the integration of velocity into the rotation and derivation into the acceleration proceeded. In many cases, the integration of seismic data was related to the visible bias generation in the rotation rate. In such a situation, additional filtering with forcing zero phases is usually utilized.

In the case of the transformation of rotational velocity into rotational acceleration, there was no need for additional filtering implementation. As a result, some differences between $\Delta_{v}$ and $\sqrt{\Delta_{a}}$ rate may be observed.

### 3.2. Field Measured Rotation in the Function of Seismic Energy Emitted

The effect of seismic events energy on the field-recorded rotational displacements at the surface seismic station was considered for the example of two seismic events characterized by opposite values of the dominant frequency of a vibration. One tremor was characterized by the frequency of 2.4 Hz, while the second event reached a dominant frequency of 10 Hz. Details about both seismic events are presented in Table 1. Based on the field 6-DoF measurements, the components of that R-wave plane were denoted as follows:(18)$$u_{x}=A_{1}0.05026\left[\exp\left(-0.04447z\right)-0.5667\exp\left(-0.01815z\right)\right]\sin\left[0.05026\left(x-v_{R}t\right)\right]\;$$
(19)$$w_{z}=A_{2}0.05026[\left(-0.8847\exp\left(-0.04447z\right)+1.5641\exp\left(-0.01815z\right)\right]\cos\left[0.05026\left(x-v_{R}t\right)\right]$$

The maximum value of the rotation *φ*_y_ on the surface (*z* = 0) may therefore be calculated from the following equations:(20)$$\phi_{y}=\phi_{y,w}=-0.00172\cdot \left(A_{1}+A_{2}\right)$$
where
(21)$$\phi_{y,w}=\frac{\partial w_{z}}{\partial x}=-A_{2}{\left(0.05026\right)}^{2}\left(1.5641-0.8847\right)=-A_{2}\cdot 0.00172\;rad$$

However, the field measurements (Table 1) indicate that the resultant induced rotation reaches a value at most equal to about
$$\phi_{o}=\sqrt{R_{x}^{2}+R_{y}^{2}\;}=\sqrt{0.002^{2}+0.003^{2}}=\pm 0.0036\;mrad$$

Assuming $\phi_{y}=\phi_{0}$ allows calibrating the multiplier (*A_1_ + A_2_*) from the relationship
(22)$$A_{2}=\frac{\pm 0.0036}{1000\times 0.00172}=\pm 0.0021$$

In general, assuming the linear dependency between any dissipated seismic energy *E_s_* and the multiplier (*A_1_ + A_2_*), one may specify the amplitudes of the R–wave components adequately using the following equation:(23)$$A_{2}=\frac{0.0021\cdot E_{s}}{1.1E7}=1.91\times 10^{-10}E_{s}$$
which is valid only for the prefixed distance between the event occurrence hypocenter and the location of the measurement station (Figure 13).

Taking into account Equations (21) and (22), one may also expect at the seismic station the rotational effect $R_{y}$ due to the seismic event of different energy *E_s_* that theoretically occur close to the focal location of recorded event No. 1:(24)$$R_{y,s}=3.28\times 10^{-13}\cdot E_{s}\;[\mathrm{rad}]$$

Considering tremor No. 2 (Table 1) of the dominant frequency *f* = 10 Hz, one may obtain relationships similar to Equations (23) and (24); however, involving different coefficients:(25)$$A_{2}=\frac{8.48E-6\cdot E_{s}}{9.7E5}=8.746\times 10^{-12}E_{s}$$
(26)$$R_{y,s}=2.61\times 10^{-13}\cdot E_{s}\;[\mathrm{rad}]$$

The results of the calculations of theh dependence between the angle of rotation, the energy of the seismic event, its dominant frequency and multiplier *A_2_* are presented in Figure 13.

## 4. Discussion

Assuming that the damping decrement does not depend upon the direction of the R-wave’s propagation, the results obtained above may exclusively deal with tremors, which occurred at the mentioned above distances from the seismic measurement station. However, due to possible differences in rock mass geological structures along the way of the wave transmissions, it is more justified to say that the presented analyses concern the rotational effects of these seismic events, which occurred or will occur in the distance of about 6 km from the 6-DoF measuring post.

With such conditions, one can say that the results of the calculations presented above show that the risk of the R wave for the stability of the structure on the surface is closely related to the amount of energy emitted from the seismic phenomenon, the distance at which it occurred and the value of the dominant frequency of the seismic wave itself. It can also be seen that the lower the dominant frequency of vibrations, the greater the effect of the wave in the range of displacements of the rotational nature. At the same time, it is worth paying attention to the visible in Figure 12 lower limit value f_min_ ≈ 2.3 Hz of the dominant frequency recorded (induced) seismic events, which, in turn, only incidentally exceeds the upper limit f_max_ ≈ 10 Hz.

Thus, the most serious effects of the R wave can be expected when its dominant frequency is close to the value of 2.3 Hz and the energy reaches *E_s_* = 2.5 × 10^9^ J, i.e., the highest energy value recorded ever in the Lowers Silesian Copper basin, Poland [53]. In this case, the value of A_2_ recorded at the seismic station was equal to about 0.477, while the rotation calculated from Formula (24) exceeded 0.000819 rad (0.047 deg). On the Richter scale, the magnitude of such mining tremors ranged from 3.8 to 4.0. In an extreme case, when the energy *E_s_* ≈ 1.0 × 10^10^ J, the rotational effect on the surface range value *R_y_* = 0.188 deg (*A_2_* = 1.9). Based on the analyzed events, one may conclude that the results of calculations correspond with the field 6-DoF measurements to a satisfying degree. Nevertheless, the reliability of the developed formulas should be improved with the further collection of field data from high-energy mining-induced tremors. Still, the models obtained herein can already be a basis to implement R-wave characteristics into numerical analyses used for stability assessment/analyses of surface objects subjected to rotational waves induced in the LSCB region.

## 5. Conclusions

This paper presents the theoretical models describing Rayleigh surface waves, characteristic for the near-field to the seismic source, and field measurements based on verified and validated information. The results of the analytical solution were compared to data collected with a 6-DoF seismic monitoring system equipped with three-axial rotational and three-component translational seismometers. As a result, the amount of seismic energy was related to the expected highest amplitude of the rotational vibrations by the iterative determination of the local site parameters A_1_ and A_2_, which, due to the lack of knowledge about seismic rotation, were treated in previous studies as constant values. The satisfactory compliance of the results of calculations and measurements made it pos-sible to carry out a preliminary prediction of the magnitude of the seismic rotation depending on the energy and frequency of the mining tremor as the source of seismic vibrations. Such an approach is highly innovative and has a direct impact on the possibility of using the corrected theoretical equations for numerical calculations. This, in turn, results in the possibility of performing more reliable object stability analyses with the use of six components of ground motion and not only three, as it was practised until now. For example, knowing that, in many cases, the failure of earthen dams is characterized as a rotational one with a circular shape of the slip surface, the neglecting of rotational seismic forces may lead to the under/overestimation of a real stability index with respect to the slopes subjected to dynamic load conditions.

Since long-term 6-DoF measurements of tremors induced by mining activities were only carried out in Poland, further measurements and analyses are required to determine such parameters as:the wave’s dominant frequency,the wave’s propagation velocity in local conditions, e.g., in the mining areas of Polish copper mines andattenuation of the rotational component of seismic waves in near- and far-wave fields.

Partial information of these terms is available in the literature, but according to the authors’ analyses, it is necessary to set up a wide 6-DoF seismic network at the regions of the mining-induced seismicity which will positively affect the current knowledge about seismic wave propagation in near-field conditions. This will be also be the basis to making quatitative risk assessments more realistic than those prepared so far.

## Figures and Tables

**Figure 1 sensors-21-03566-f001:**
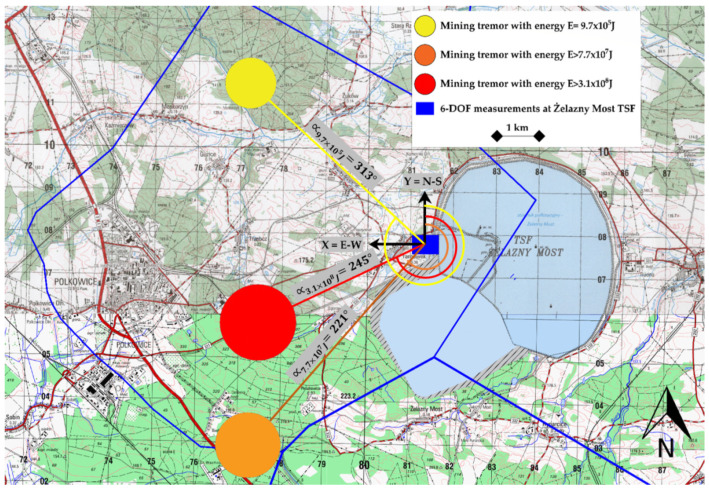
Location of the 6-DOF measuring station and epicenters of the analyzed mining tremors in the LSCB area.

**Figure 2 sensors-21-03566-f002:**
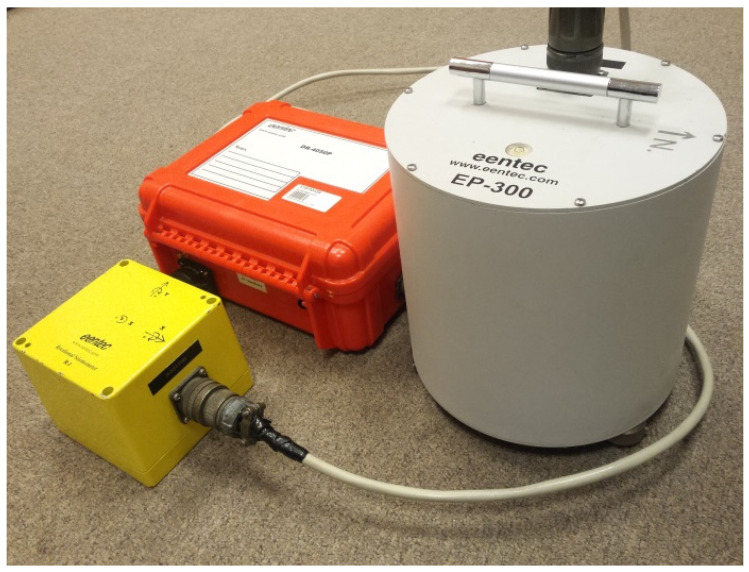
System for 6-DOF measurements installed in the Rudna mining region.

**Figure 3 sensors-21-03566-f003:**
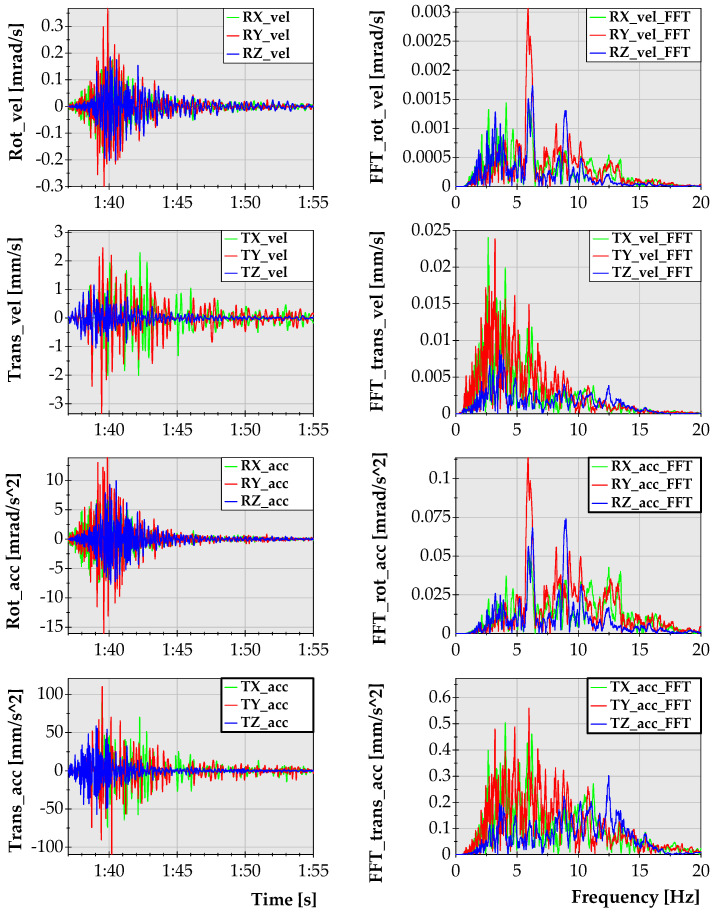
Amplitude (**left**) and frequency (**right**) characteristics of the velocity and acceleration waveforms generated by the tremor of energy of E = 3.1 × 10^8^ J that occurred at a distance of 4446 m from the seismic station.

**Figure 4 sensors-21-03566-f004:**
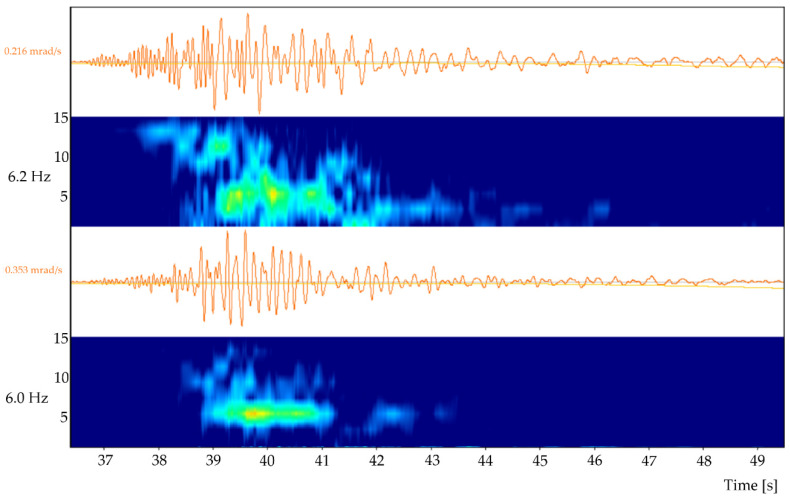
Time–frequency decomposition of the horizontal components of the rotational velocity records of the mining tremors for the energy of E = 3.1 × 10^8^ J.

**Figure 5 sensors-21-03566-f005:**
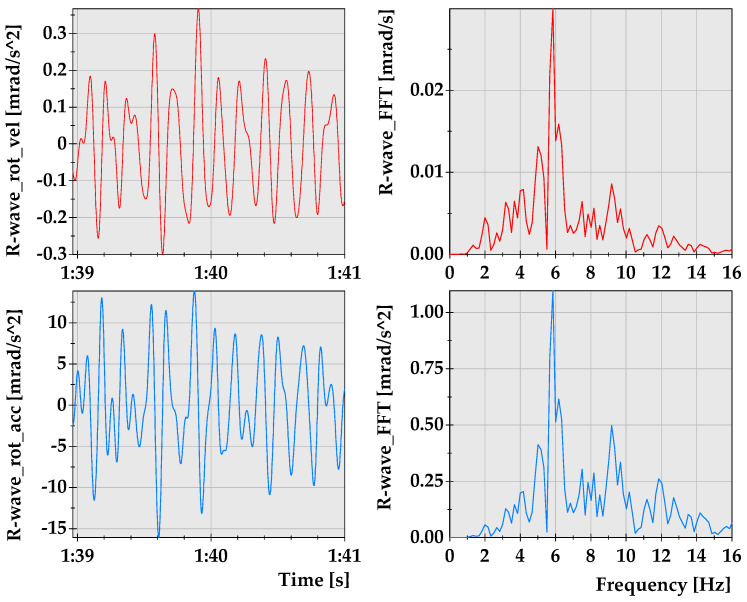
Velocity and acceleration characteristics of the R-wave generated by the mining tremor for the energy of 3.1 × 10^8^ J.

**Figure 6 sensors-21-03566-f006:**
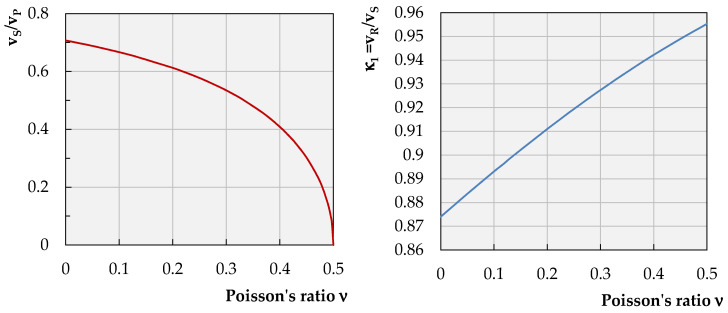
Relationship between the Poisson’s ratio values and velocities of waves R, P and S.

**Figure 7 sensors-21-03566-f007:**
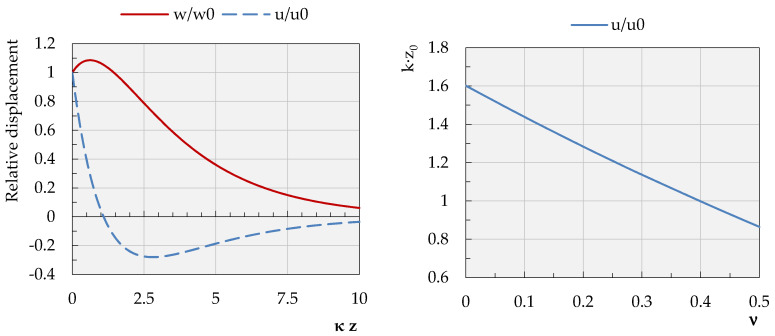
Changes in the values of the relative displacements (**left**) and normalized with respect to the *κ*, attenuation depth of the horizontal component of the wave R displacement in the function of Poisson’s ratio ν.

**Figure 8 sensors-21-03566-f008:**
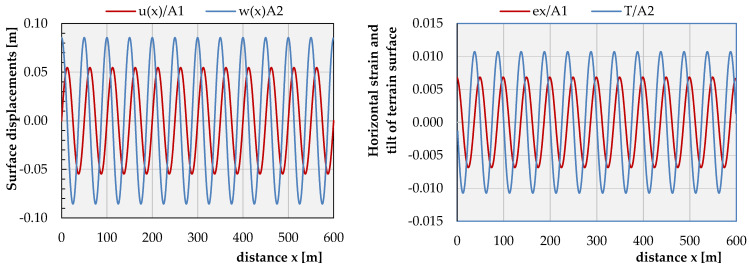
Instantaneous amplitudes of the surface vibration induced by the R-wave (**left**) and instantaneous amplitudes of the surface horizontal strain and the surface rotation around the y-axis (**right**).

**Figure 9 sensors-21-03566-f009:**
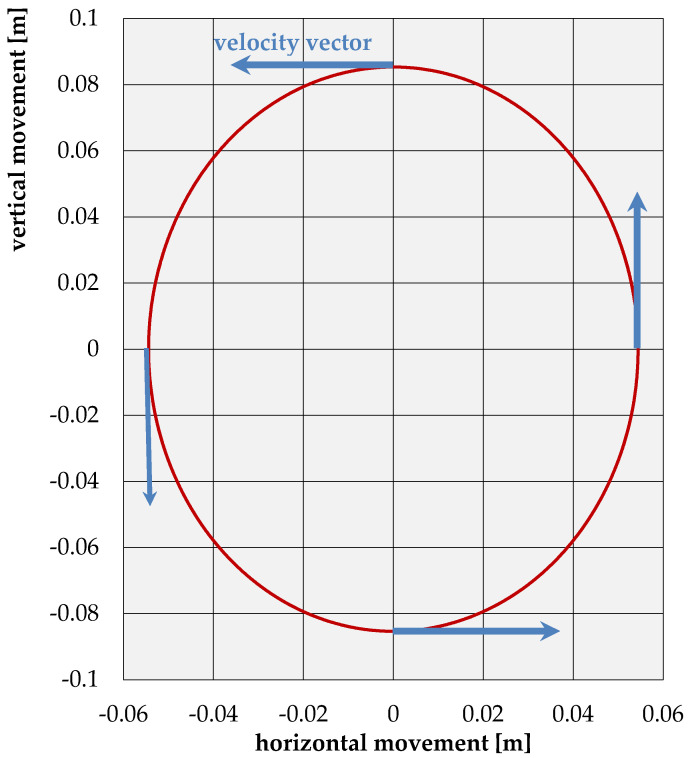
Displacement trajectories of the points located on the surface of an elastic half-space for the case under consideration (*ν* = 1/3).

**Figure 10 sensors-21-03566-f010:**
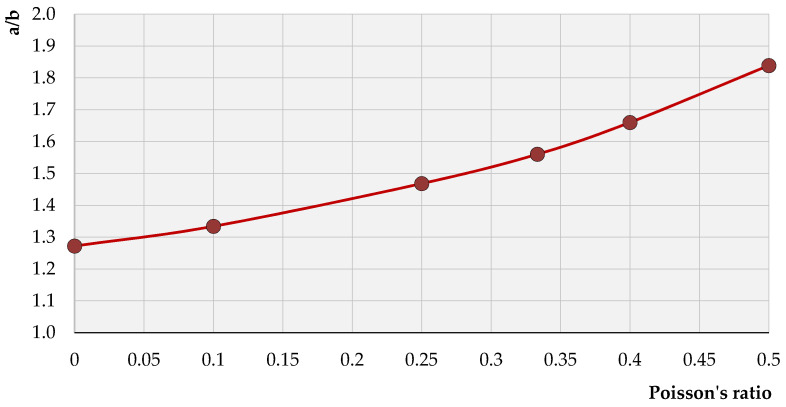
The ratio of the major to the minor axes of the ellipse in the function of Poisson’s ratio (*ν*).

**Figure 11 sensors-21-03566-f011:**
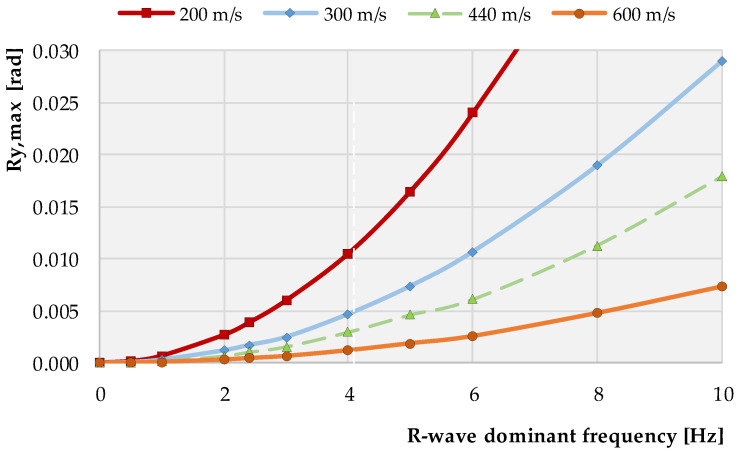
Relationship between the R-wave dominant frequency, its velocity and the maximum amplitude of the *Ry* rotation around the y-axis ($\frac{\partial w}{\partial x}=T_{y}=R_{y})$, calculated based on Equation (14).

**Figure 12 sensors-21-03566-f012:**
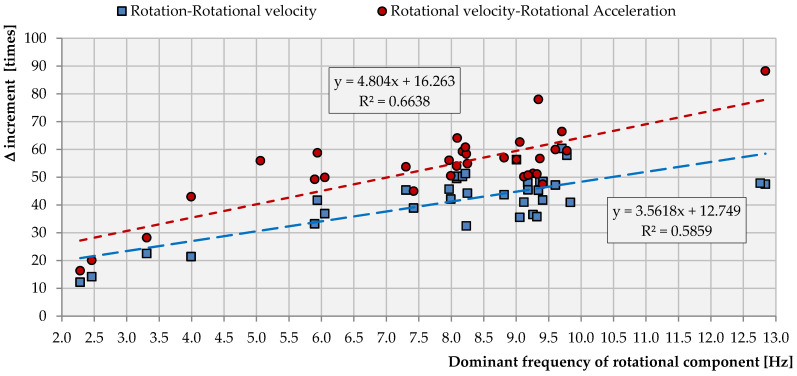
The relation between rotation, rotational velocity and rotational acceleration.

**Figure 13 sensors-21-03566-f013:**
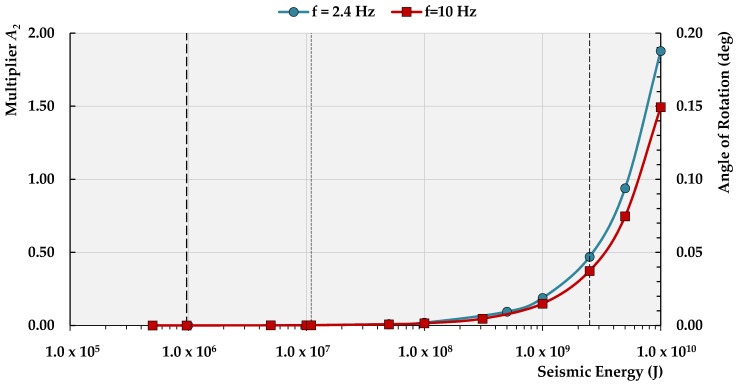
The relation between the seismic energy *E* and the multiplier *A_2_* values for the events that occurred at a distance of about 6000 m from the seismic measurement station.

**Table 1 sensors-21-03566-t001:** Basic data concerning the rotation induced by selected mining tremors recorded at the seismic station located at Zelazny Most Tailing Pond.

No.	Date	Domin. Frequency [Hz]	Energy [J]	Distance to Hypocenter [m]	The Maximum Absolute Value of Rotation ϕ [mrad]		
					R_X_	R_Y_	R_Z_
1	2019-01-12	2.4	1.1 × 10^7^	6372	1.8 × 10^−3^	2.9 × 10^−3^	1.7 × 10^−3^
2	2019-01-29	6.0	3.1 × 10^8^	4446	6.9 × 10^−3^	1.0 × 10^−2^	5.6 × 10^−3^
3	2019-07-04	10	9.7 × 10^5^	5722	2.4 × 10^−4^	8.8 × 10^−5^	6.0 × 10^−6^

**Table 2 sensors-21-03566-t002:** Typical values of the propagation velocities of the longitudinal and transverse waves in selected geotechnical media (based on [50]).

Type of Material	$v_{P}$ [m/s]	$v_{S}$ [m/s]
Aluvium, river sediments	500–2100	$v_{S}=v_{P}\sqrt{\frac{1-2\nu }{2\left(1-\nu \right)}}$
Clays	1100–2500	
Sands	200–2000	
Glacial deposits	400–1700	
Sandstones	1400–4500	
Shales	2300–4700	
Limestone soft, coherent, recristallized	1700–4200; 2800–6400; 5700–6400	
Dolomite	3500–6900	
Granite, Granodiorite	4600–6000	2800–3200
Diabase	5800–6000	
Gabro	6400–6700	3400–3600
Basalt	5400–6400	2700–3200
Metamorfic shales	4200–4900	2500–3200
Gneisses	3500–7500	3300–3700
Water	1450	-
Air	335	-

**Table 3 sensors-21-03566-t003:** Values of the Poisson’s ratio and *κ*_1_ ratio for different kinds of soils (based on [51]).

	**Noncohesive Soils**			**Cohesive Soils**			
Type of soil	Gravel	Coarse and medium sands	Fine and dusty sands	Consolidated moraine loams	Other cohesive consolidated soils and non-consolidated moraine cohesive soils	Other non-consolidated cohesive soils	Clay
Poisson’s ratio	0.20	0.25	0.30	0.25	0.29	0.32	0.37
Ratio *κ*_1_	0.9110	0.9194	0.9274	0.9194	0.9258	0.9305	0.9380
	SULPHATE ROCKS		CARBONATE ROCKS			SANDSTONES	
Type of rocks	Massive anhydrite	Gypsum-anhydrite	Dolomitic limestone	Calcareous Dolomites	Dolomites	Quartz with a carbonate binder	Quartz with a clay binder
Poisson’s ratio	0.26	0.21	0.24	0.25	0.23	0.20	0.13
Ratio *κ*_1_	0.9210	0.9127	0.9178	0.9194	0.9161	0.9110	0.8986

## Data Availability

The data presented in this study are available on request from the corresponding author. The data are not publicly available due to internal KGHM CUPRUM regulations.

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
