# Peer review of "Validation of Rayleigh Wave Theoretical Formulation with Single-Station Rotational Records of Mine Tremors in Lower Silesian Copper Basin"

_sensors, 2021, doi:10.3390/s21103566_

Round 1

Reviewer 1 Report

The paper investigates an interestic application of the theoretical notion of Rayleigh wave characteristic for the Legnica GÅ‚ogów Copper Basin area. The paper shows the improved the direct measurement-based numerical slope stability calculations. Methodology is appropriate and the structure of the paper well balanced. English is also good. There are some minor revisions I suggest: 1. The introduction needs to state clearly the novelties of the paper in order to understand that it is not only an application. 2. Formulas 1-7 is nothing new. This part needs to be reduced. 3. The reason for chosing a plane state for the preliminary analysis needs to be explained. 4. Figure 8 shows a continue function as an analytical formulation. Is it correct or points needs to be plotted? 5. Conclusions They need to be more focused on the novelties and how the paper develops the gap in literature. For example Nobody needs that "the article presents the classical knowledge of Rayleigh surface waves in terms of their theoretical record" For these reasons, I recommend minor revisions before being accepted

Author Response

We would like to thank the Reviewer for the effort and detailed review. We appreciate all the comments and finally, we have incorporated all of the suggestions made by the reviewer. We believe that changes introduced according to reviewer comments and suggestion will enrich the scientific quality of our paper. All changes are highlighted in the corrected manuscript (any changes are visible in Track Changes mode in Microsoft Word). Moreover, our feedback and responses may be found in the present document

Comment 1. The introduction needs to state clearly the novelties of the paper to understand that it is not only an application.

Ad. 1. The introduction has been rewritten according to the reviewer suggestion. The scope and novelty of the presented solution have been highlighted in the last paragraph of the introduction.

Comment 2.  Formulas 1-7 is nothing new. This part needs to be reduced.

Ad. 2.We agree with the Reviewer. The text and formulas description has been changed accordingly

Comment 3.  The reason for choosing a plane state for the preliminary analysis needs to be explained.

Ad. 3. According to the reviewer suggestions, the reasons have been explained at the beginning of subsection 3.1. Rayleigh wave mathematical form assessment

Comment 4. Figure 8 shows a continued function as an analytical formulation. Is it correct or points needs to be plotted?

Ad. 4.Figure 8 have been plotted based on points. We have added these points to the graph.

Comment 5. Conclusions They need to be more focused on the novelties and how the paper develops the gap in the literature. For example, Nobody needs that "the article presents the classical knowledge of Rayleigh surface waves in terms of their theoretical record"

Ad. 5.  We agree with the reviewer suggestion. The conclusion section has been rewritten, expanded according to this comment.

Reviewer 2 Report

The article presents the classical knowledge of Rayleigh surface waves in terms of their theoretical record and the resulting properties of this wave type accompanying induced seismic phenomena.

The paper is good and clearly written I have only few small recommendations (minor revision): 

  1. The objective and the novelty of the paper have to be clearly stated in the Introduction part of the article. The objective seems to be clear (lines 77-84), but the reviewer strongly recommend to emphasise what the novelty of the paper is.
  2. The abstract lacks conclusions.
  3. Lines 44-47: “According to the International Commission on Large Dams (ICOLD) report in the XXIst century, at least once a year the disaster-related to the loss of the stability of TSF slopes have been observed.” Please add reference(s).
  4. Lines 115-117: “According to the authors' knowledge and literature review, there is no available FEM-based numerical code, which allows implementing a rotational component of the seismic wave into the calculation.” That is not true, for example the Abaqus software allows introducing rotational components. Please remove this comment.
  5. Lines 161-163: “However, these waves additionally produce at the surface another type of wave which travel only over the surface, the little penetrating interior of the body. They are called surface waves from which the Rayleigh wave is one of the most important from point of view of damage severity caused by the natural and induced seismicity.” Please remove this information since it is trivial and obvious for engineering community interested in seismic issues.
  6.  The language needs to be polished.

Author Response

We would like to thank the Reviewer for the effort and detailed evaluation of our manuscript. We appreciate all the comments. Most of the changes suggested by the reviewer were made accordingly. Those changes are highlighted in the corrected manuscript (any changes are visible in Track Changes mode in Microsoft Word). Moreover, our feedback and responses may be found in the present document

Comment 1. The abstract lacks conclusions.

Ad. 1. We made the required changes according to the reviewer suggestion. The information about conclusions has been added to the abstract.

Comment 2. Lines 44-47: “According to the International Commission on Large Dams (ICOLD) report in the XXIst century, at least once a year the disaster-related to the loss of the stability of TSF slopes have been observed.” Please add reference(s).

Ad. 2. According to the reviewer comment, two references were added (25-26)

Comment 3. Lines 115-117: “According to the authors' knowledge and literature review, there is no available FEM-based numerical code, which allows implementing a rotational component of the seismic wave into the calculation.” That is not true, for example, the Abaqus software allows introducing rotational components. Please remove this comment.

Ad. 3. Besides the literature review, authors have contacted also software developers and resellers with the question about the possibility of implementing seismic rotation into caluations. We have contacted the support of MIDAS GTS NX, RocScience, Geostudio, and Abaqus. All expert has responded that there is no “build in” procedure of implementing seismic rotation into the calculation. All seismic load characteristic is presented in a translational manner in form of displacement, velocity or acceleration. A rotational speed which is available to determine in each software describes the rotation of the element around its axis e.g. rotation of bar elements. At the same time, there is no possibility to determine the rotational velocity of volumetric structures generated by vibrations.

Comment 4. Lines 161-163: “However, these waves additionally produce at the surface another type of wave which travel only over the surface, the little penetrating interior of the body. They are called surface waves from which the Rayleigh wave is one of the most important from point of view of damage severity caused by the natural and induced seismicity.” Please remove this information since it is trivial and obvious for the engineering community interested in seismic issues.

Ad. 4. According to the reviewer comment, the sentence in lines 161-163 have been deleted

Comment 5. The language needs to be polished.

Ad. 5. Minor corrections have been made according to the reviewer comment.

Reviewer 3 Report

The manuscript sensors-1159419 presents the study of the Rayleigh surface wave in terms of its theoretical notation and resulting properties associated with induced seismic phenomena. The parameters governing the mathematical notation of the Rayleigh wave were elaborated deeply based on the similarity to records obtained during the induced seismicity near-field monitored at measuring stations equipped with three-component rotational seismometers located in the Zelazny Most mining area (SW Poland).

I think that the manuscript is very interesting, and it can be useful for all the scientific community involved in Seismology and specifically in the study of the surface waves. Anyway, after having carefully read the manuscript, I think it can be accepted with minor revisions.

Main comments

The introduction paragraph is too long and redundant, so I suggest shortening it.

Conclusion paragraph: reword the conclusions with the bullet point approach evidencing the novelty of you work with respect to the recent results obtained in the study of the Rayleigh wave.

I suggest checking the English text by a mother tongue speaker.

Author Response

We would like to thank the Reviewer for the effort and detailed evaluation of our manuscript. We appreciate all the comments. Most of the changes suggested by the reviewer were made accordingly. Those changes are highlighted in the corrected manuscript (any changes are visible in Track Changes mode in Microsoft Word). Moreover, our feedback and responses may be found in the present document

Comment 1. The introduction paragraph is too long and redundant, so I suggest shortening it.

Ad. 1. According to the reviewer comment, the introduction have been shortened. Some information about the Newmark method has been deleted or rewritten. In the author's opinion, the rest of the introduction, concerning the issue of seismicity, lack of rotational measurements and paper purpose have been slightly changed.

Comment 2. Conclusion paragraph: reword the conclusions with the bullet point approach evidencing the novelty of you work with respect to the recent results obtained in the study of the Rayleigh wave.

Ad. 2. According to the reviewer comment, the conclusion section has been rewritten, and information about the novelty of the proposed solution has been highlighted. We would like to highlight that in our opinion bullet point approach is not the best one in the case of our article, and we have presented conclusions mostly in form of continuous text. Still, if this form is not acceptable in the reviewer opinion we may change it accordingly.

Comment 3. I suggest checking the English text by a mother tongue speaker.

Ad. 3. Minor corrections have been made according to the reviewer comment.

Round 2

Reviewer 1 Report

The paper has been improved and it is ready for pubblication

Author Response

We would like to thank the Reviewer for the effort and detailed review. We appreciate all the comments and finally, we have incorporated all of the suggestions made by the reviewer. We believe that changes introduced according to reviewer comments and suggestion will enrich the scientific quality of our paper. All changes are highlighted in the corrected manuscript (any changes are visible in Track Changes mode in Microsoft Word).

We hope that you will be satisfied with our answers and the changes we have made to the manuscript. We believe that the content of our paper in the present form is clear and will be added value to current knowledge about rotational seismology, especially in the case of near-field measuremet.

Best Regards,

Authors. 

Reviewer 3 Report

I read carefully the R1 version of the manuscript and I think it can be accepted for publication in its present form

Author Response

(The authors gave the same response as above.)
